# X-Reasoner: Towards Generalizable Reasoning Across Modalities and Domains

## Abstract

Recent proprietary models (e.g., o3) have begun to demonstrate strong multimodal reasoning capabilities. Yet, most existing open-source research concentrates on training text-only reasoning models, with evaluations limited to mainly mathematical and general-domain tasks. Therefore, it remains unclear how to effectively extend reasoning capabilities beyond text input and general domains. This paper explores a fundamental research question: *Is reasoning generalizable across modalities and domains?* Our findings support an affirmative answer: *General-domain text-based post-training can enable such strong generalizable reasoning, which is even more effective than in-domain multimodal training.* Leveraging this finding, we introduce X-Reasoner, a vision-language model with reasoning post training solely from general-domain text for generalizable reasoning, using a two-stage approach: an initial supervised fine-tuning phase with distilled long chain-of-thoughts, followed by reinforcement learning with verifiable rewards. Experiments show that X-Reasoner successfully transfers reasoning capabilities to both multimodal and out-of-domain settings, outperforming prior models trained with in-domain and multimodal data across various general and medical benchmarks (Figure 1). Additionally, we find that X-Reasoner's performance in specialized domains can be further enhanced through continued training on domain-specific text-only data. Building upon this, we introduce X-Reasoner-Med, a medical-specialized variant that achieves SOTA (state-of-the-art)-level performance on numerous text-only and multimodal medical benchmarks.

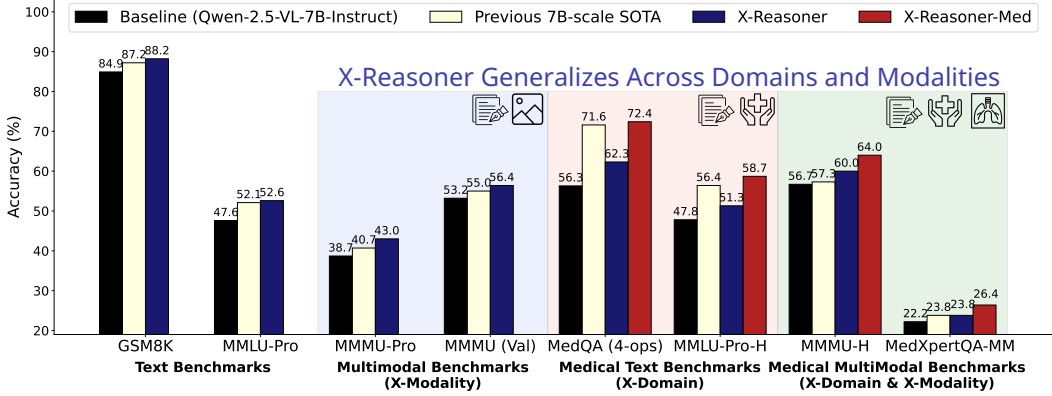

Figure 1: X-Reasoner (■ blue bars), fine-tuned solely on general domain text 📝, shows strong generalization across both modalities (e.g., multimodality 📝🖼) and domains (e.g., medicine 🩺🫁), surpassing prior models (Table 7) trained with in-domain multimodal data. X-Reasoner-Med (■ red bars), its medical-specialized variant, achieves SOTA-level performance on numerous medical benchmarks.

# 1 INTRODUCTION

Reasoning has emerged as a foundational capability in language models, paving the way for a new paradigm known as test-time scaling (OpenAI, 2024). Recent proprietary models, such as o3 (OpenAI, 2025), have begun to demonstrate strong multimodal reasoning capabilities. However, existing open-source research primarily focuses on advancing text-only reasoning, employing post-training techniques such as long CoT (Chain-of-Thought) distillation and reinforcement learning with verifiable rewards (DeepSeek-AI et al., 2025; Hu et al., 2025). More recently, a growing body of open-source work has started addressing multimodal reasoning, i.e., extending reasoning to inputs that include both vision and language (Wang et al., 2025a; Meng et al., 2025; Guo et al., 2024). Yet, these multimodal approaches often rely heavily on curating multimodal datasets which are tailored to specific tasks or domains, limiting their generalisability. In parallel, researchers have also investigated domain-specific reasoning, particularly in medicine (Zhang et al., 2025a; Lai et al., 2025; Pan et al., 2025; Su et al., 2025), where models are typically trained on narrowly scoped in-domain data with little capacity to generalize beyond their domain. As a result, how to develop models with reasoning capabilities that generalize across both domains and modalities remains an open question.

In this work, we ask a fundamental, yet under-explored question: *Is reasoning generalizable across modalities and domains?* Specifically, we investigate whether such generalizable reasoning can be achieved through general-domain text-based reasoning post-training. Beyond scientific significance, this question is also motivated by the practical advantages of general domain text-only training: i.e. its compute efficiency and the abundance and verifiability of general-domain textual reasoning data, which together allow us to avoid the cost and complexity of curating multimodal or domain-specific data. Our hypothesis is that text-based post-training, when carefully designed, can impart universal reasoning patterns that robustly transfer across both unseen domains and input modalities.

To investigate our hypothesis, we conduct an extensive empirical study using a two-stage text-only post-training recipe: supervised fine-tuning (SFT) on general-domain text data with distilled long CoTs, followed by reinforcement learning with verifiable rewards (RLVR) using mathematical textual questions. Remarkably, this pure general-domain textual training regimen proves sufficient to instill strong reasoning capabilities, enabling high performance not only on general-domain text-based tasks but also on complex multimodal and domain-specific tasks.. This finding suggests that the core structure of reasoning can indeed be acquired from general-domain text alone.

Building upon these insights, we introduce **X-REASONER**, a 7B dense vision-language model post-trained with the proposed recipe. Despite being trained only for text-based reasoning, X-REASONER achieves superior performance on a suite of both text-only and multimodal reasoning benchmarks, outperforming prior state-of-the-art 7B models that were explicitly trained with multimodal reasoning supervision, on challenging tasks including MMMU, MMMU-Pro and MathVista. We further demonstrate that the improvement from X-REASONER can generalize to specialized domains such as medicine. To boost in-domain performance even further, we explore the impact of incorporating domain-specific textual data. To this end, we introduce **X-REASONER-MED**, a medical-specialized variant of X-REASONER, which undergoes additional post-training on medical domain text. X-REASONER-MED sets new records on numerous textual and multimodal medical tasks.

Our key contributions can be summarized as the following:

1. **Generalizable Reasoning Study:**
   - We conduct an in-depth study and find that training reasoning solely on general-domain text yields the most effective generalization across both input modalities and domains.
   - We find that combining SFT and RL yields the strongest gains, and that math text data serves as better generalization anchor than domain-specific multimodal data in RL.

2. **Introduction of X-REASONER:**
   - We propose an effective post-training recipe for vision-language reasoning models that relies entirely on general-domain text-based data.
   - X-REASONER improves performance across modalities and domains, remarkably, outperforming models trained on multimodal data.

3. **Medical Domain Extension via X-REASONER-MED:**
   - We present X-REASONER-MED, a domain-adapted variant of X-REASONER trained on medical text, setting new 7B-scale records across medical text-only and multimodal tasks.

## 2 A JOURNEY TOWARDS GENERALIZABLE REASONING

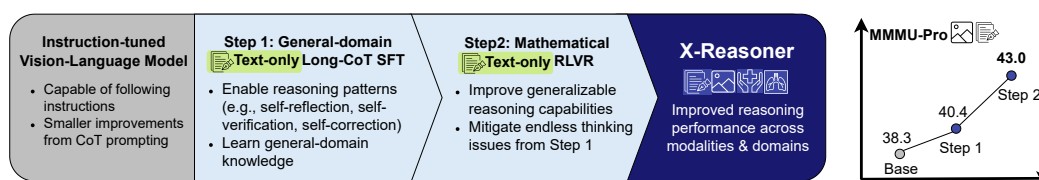

Figure 2: (Left) Our recipe for generalizable reasoning: based on an instruction-tuned VLM, we first conduct SFT on general-domain text data with distilled long CoTs. This is followed by RLVR on mathematical textual questions. This resulting model, X-REASONER, exhibits significantly enhanced reasoning capabilities across modalities and domains. (Right) Model performance on MMMU-Pro (multimodal task) steadily improves at each stage of our recipe.

In this section, we embark on a systematic exploration to understand whether generalizable reasoning can be effectively acquired through general-domain text-based post-training. Here, we define *generalizable reasoning* as the ability to transfer reasoning capabilities not only across different task distributions but also across modalities and domains. To thoroughly address this question, we break it down into two sub-questions corresponding to two predominant post-training strategies used to foster reasoning: SFT and RL. Starting from an instruction-tuned vision-language model (VLM) capable of following instructions yet benefiting less from CoT prompting (see Section 2.1), we perform a detailed empirical study utilizing these two post-training methods. Our objective is to identify a robust and effective recipe that achieves strong, generalizable reasoning capabilities purely through general-domain text-based training. All experiments in this study are initialized from Qwen-2.5-VL-7B, which is also the baseline. Evaluation tasks used are summarized Appendix E.

### 2.1 RQ1: WILL GENERAL-DOMAIN TEXT-ONLY SFT IMPROVE GENERALIZABLE REASONING?

Our first research question investigates whether SFT on general-domain textual data can already enable generalizable reasoning, focusing on (1) the extent to which generalization can occur across domains and modalities, and (2) the essential role of reasoning in enabling such generalization.

**Extent of Generalization** To examining the extent of the generalization from SFT training, we perform SFT on Qwen-2.5-VL-7B using `OpenThoughts-114k` (Open Thoughts Team, 2025), a *text-only* dataset covering *general-domain* data including math and science long-CoT reasoning distilled from DeepSeek-R1 (DeepSeek-AI et al., 2025). Results in Table 1 show the SFT models yield consistent performance gains across the following generalization axes: cross-domain (X-domain), cross-modality (X-modality) and the intersection of the two axes (We additionally observe text-only SFT generalizes across distributions/tasks Appendix G). This indicates that SFT with general-domain text-data already shows strong generalization capabilities across modalities and domains.

**Role of CoT Reasoning in Generalization** In Table 1, we observe that reasoning is crucial at inference time (CoT prompt templates are in Appendix D). Even among the baseline models, those equipped with CoT prompting outperform those without it, underscoring that explicit reasoning is a prerequisite for high performance on reasoning-centric tasks. However, the improvement from CoT prompting in the baseline is notably smaller compared with SFT, indicating that the VLM's inherent CoT capability prior to SFT is limited and the long-CoT reasoning learned from SFT is essential to unlock the full reasoning power. To further isolate the contribution of reasoning in SFT training, we compare models trained with and without explicit CoT supervision using the same OpenThoughts data: (1) a non-CoT SFT model trained directly on input-label pairs, and (2) a CoT SFT model trained with reasoning traces. Across all experiments in Table 1, the CoT SFT model consistently outperforms its non-CoT counterpart. Moreover, the CoT SFT models not only perform better in-domain/modality but also exhibit stronger transfer capabilities. Notably, the performance lift from CoT SFT over the CoT baseline is substantially larger than the corresponding improvement from non-CoT SFT over the non-CoT baseline across all generation settings. This suggests that explicit reasoning is the key to achieving both the highest performance and the optimal generalization across new domains and modalities.

**Forced-Exiting: Mitigating Endless Thinking in Long-CoT SFT** While long-CoT SFT effectively learns transferrable reasoning patterns, such as self-reflection, verification, and correction (see qualitative analyses in Appendix N), it occasionally leads to endless thinking with non-terminating outputs. To address this, we implement a forced-exiting mechanism inspired by Muennighoff et al. (2025) to append '`</think>`' after a length threshold is reached. This mechanism effectively mitigates endless thinking and therefore improves final results (See Appendix H for details).

Table 1: Comparing baseline (Qwen-2.5-VL-7B) and SFT models trained on general-domain text-only data with and without CoT on evaluation benchmarks across modalities and domains.

| Task | Modality | Domain | Non-CoT | | CoT | |
|------|----------|--------|---------|-----|-----|-----|
| | | | Baseline | SFT | Baseline | SFT |
| MMLU-Pro | Textual | General | 39.5 | 39.5 (+0.0) | 47.6 | **50.4** (**+2.8**) |
| MedQA (X-domain) | Textual | Medical | 49.3 | 52.1 (+2.8) | 50.5 | **55.2** (**+4.7**) |
| MMMU-Pro (X-modality) | Multimodal | General | 34.6 | 36.0 (+1.4) | 38.3 | **40.4** (**+2.1**) |
| MMMU-Pro-H (X-modality&domain) | Multimodal | Medical | 28.1 | 30.2 (+2.1) | 34.3 | **37.9** (**+3.6**) |

> **Takeaway 2.1.1**
>
> SFT on general-domain text-only data, when enriched with long CoTs, can endow models with generalizable reasoning capabilities that transfer across tasks, domains, and modalities.

## 2.2 RQ2: WILL GENERAL-DOMAIN TEXT-ONLY RL IMPROVE GENERALIZABLE REASONING?

Reinforcement Learning (RL) has become an effective approach for training models to reason. In this section, we explore whether general-domain (particularly mathematical) text-only RL can promote generalizable reasoning.

**Learning Algorithm** We adopt GRPO (Shao et al., 2024) as our reinforcement learning algorithm (Sutton, 1988), which avoids value functions by computing advantages within query-specific groups, making it preferable to methods like PPO (Schulman et al., 2017) (see Appendix B). To enhance training, we apply recent advances (Liu et al., 2025a; Yu et al., 2025): (1) a higher clipping threshold to boost response diversity and prevent entropy collapse; (2) a token-level policy gradient to reduce response-length bias; and (3) minimal or no KL penalty, which benefits long CoT reasoning. For rewards, we use verifiable task accuracy—assigning 1 for semantically correct responses and 0 otherwise—instead of learned reward models, which are prone to reward hacking (Gao et al., 2022). No format rewards are needed, as the model reliably adheres to formatting.

**The Role of Training Data: Math Text as a Generalization Anchor** A central question in generalizable RL training is what type of training data best supports reasoning generalization? We hypothesize that math textual questions are particularly effective, as math tasks naturally elicit long, structured chains of thought that should benefit transfer across domains and modalities. To test this, we finetune Qwen-2.5-VL-7B with RL on math textual questions from Orz (Hu et al., 2025). We compare this approach with RL trained with popular existing in-domain and multimodal training data including (1) MedQA, a domain-specific medical textual QA dataset . (2) ThinkLite (Wang et al., 2025b), a large-scale curated multimodal general-domain VQA dataset. and (3). OmniMed-VQA, a large-scale multimodal medical-domain VQA dataset (We follow previous studies Pan et al. (2025) to split 80% and 20% of the data as the train and test set). We report results across all evaluation combinations of domain and modality: {general, medical} × {text, multimodal}. For additional reference, we include in-distribution test set results for RL datasets that have corresponding test splits. Specifically, we report results on MathVista testmini, MedQA test and OmniMedVQA test for ThinkLite RL, MedQA RL and OmniMedVQA RL. Results in Table 2 reveal RL trained with math text data achieves the overall best generalization with the highest average performance. Notably, math text RL can achieve better performance in MMMU-Pro (a general-domain multimodal task) than RL trained on general-domain multimodal data (ThinkLite). Similarly, on MMMU-Pro-Health, a medical multimodal task, math text RL outperforms OmniMedVQA RL. We observe that while OmniMedVQA does achieve near-perfect performance on its own test set: 97% accuracy, it fails to generalize to other tasks—even those that are in-domain and multimodal. These findings highlight that math text data offers the most effective foundation for cross-domain and cross-modality generalization, outperforming in-domain multimodal training.

Table 2: Comparing the effect from different training data on RL generalizatio from Qwen-2.5-VL-7B. For some RL data, in-distribution (in-distr.) test results are also shown. Each result reports average accuracy over 5 runs (SD. in Table 15). gen.=general; med.=medical; OMV=OmniMedVQA.

| RL Data | Domain | Modality | GSM8K (gen. text) | MMMU-Pro (gen. multimodal) | MMLU-Pro-H (med. text) | MMMU-Pro-H (med. multimodal) | Average (all) | In-distr. |
|---------|--------|----------|-------------------|---------------------------|------------------------|------------------------------|---------------|-----------|
| Baseline | - | - | 86.0 | 38.3 | 47.8 | 34.3 | 51.6 | - |
| math Orz | general | text | **89.0** | **41.2** | 51.8 | 36.9 | **54.7** | - |
| MedQA | medical | text | 86.9 | 39.7 | **53.9** | 34.4 | 53.7 | |
| ThinkLite | general | multimodal | 87.4 | 40.1 | 49.8 | **37.5** | 53.7 | 73.0 |
| OMV | medical | multimodal | 85.1 | 36.2 | 45.3 | 29.0 | 48.9 | 97.0 |

> **Takeaway 2.2.1**
>
> Math text data provides effective generalization anchor for cross-domain and cross-modality reasoning in RL, outperforming even in-domain multimodal training.

**RL vs SFT** Prior work has positioned RL as a powerful yet volatile method for promoting generalizable reasoning (Chu et al., 2025), while SFT is widely recognized for its stability and efficacy in capturing structured reasoning patterns. In addition, RL is often plagued by training instability and convergence issues, particularly when applied in isolation (Yeo et al., 2025). To better understand their relative strengths, we conduct comparison between RL and SFT in fostering generalization across domain shifts and modality transitions. Specifically, we compare general-domain text-only SFT (trained on general-domain OpenThoughts data) and RL (trained on math questions from Orz), and their combination (SFT + RL). Alongside the general-domain text task (MMLU-Pro), we evaluated the models in three distinct generalization settings: cross-domain (medical text: MMLU-Pro-Health), cross-modality (multimodal: MMMU-Pro), cross domain&modality (multimodal medical: NEJM Image Challenge). In Table 3, we first observe that general-domain SFT and RL can both significantly improve from baseline across all the generalization settings, indicating both SFT and RL can elicit generalizable reasoning. We further notice that pure RL is overall slightly better than SFT. Interestingly, there is a synergy effect when we apply RL after SFT. We hypothesize that it is because RL can benefit from the long and structured reasoning foundation established by SFT, and further refines the model's capabilities via RLVR. This is supported by the observation that SFT + RL maintains the long response length induced by SFT. We also observe that SFT + RL is able to regulate the endless thinking issue from SFT as the percentage of responses that exceeds the maximum length decreases during RL training, as shown in the training response clip ratio in Figure 5. In summary, the hybrid approach, general-domain text-only SFT + RL, achieves the best results across all generalization settings, effectively combining SFT's stability and inductive strength with RL's reward-guided optimization[1].

Table 3: Comparison of general-domain text-based RL, SFT, and SFT + RL in general-domain textual task (MMLU-Pro) and generalization setups including cross-modality (X-modality), cross-domain (X-domain) and combined cross-domain&modality (X-modality&domain) tasks. SFT is trained with OpenThoughts data and RL is trained with math data. Baseline is Qwen-2.5-VL-7B

| Method | MMLU-Pro (G-domain text) | MMLU-Pro-Health (X-domain) | MMMU-Pro (X-modality) | NEJM Image Challenge (X-domain&modality) | Response Len (#words) |
|--------|--------------------------|----------------------------|------------------------|-------------------------------------------|------------------------|
| Baseline | 47.6 | 47.8 | 38.3 | 41.8 | 122 |
| SFT | 50.4 | 50.4 | 40.4 | 45.0 | 946 |
| RL | 52.3 | 51.8 | 41.3 | 45.0 | 461 |
| SFT + RL | **53.3** | **53.1** | **42.5** | **45.7** | 977 |

> **Takeaway 2.2.2**
>
> Combining SFT with RL, both trained on general-domain text-only data, proves to be the most effective strategy for achieving optimal performance and robust generalization.

---

[1]We also conduct a more controlled comparison between SFT and RL, both trained on the same MedQA data, and arrive at the same conclusion (Table 10).

## 2.3 Putting it all together: X-Reasoner

Our investigation reveals that *general-domain text-only post-training, when carefully designed, can drive strong generalizable reasoning across tasks, domains, and modalities.* Therefore, we conclude our investigation by consolidating our findings into a coherent training recipe, illustrated in Figure 2, culminating in a powerful generalizable reasoning model named **X-Reasoner**. Specifically, initialized with `Qwen2.5-VL-7B-Instruct` (Bai et al., 2025) (Table 19 shows X-Reasoner can be extended to additional sizes and model family), our training recipe follows:

**Step 1: text-only general-domain Long-CoT SFT**. We begin with SFT to elicit explicit structured reasoning using long-CoT traces. The train data is the general-domain `open-thoughts/OpenThoughts-114k` dataset (Open Thoughts Team, 2025), which contains reasoning traces on math, coding and science questions, distilled by DeepSeek-R1 (DeepSeek-AI et al., 2025) (We also compared with using math data from RL stage for SFT but found worse performance in Appendix J.) We performed SFT for 4 epochs with a learning rate of $1 \times 10^{-5}$.

**Step 2: text-only math RLVR**. After SFT, we further refine our model using RL with verifiable rewards to enhance its reasoning accuracy and generalization. We train our model on `Orz-math-57k`, a set of 57k mathematical textual questions curated by Hu et al. (2025). During this stage, we set the total training episodes/epochs to 3, use a learning rate of $3 \times 10^{-6}$, a global batch size of 128, and sample 8 rollouts per query with a maximum response length of 4,096 tokens.

SFT used 8×40GB A100 GPUs for 8 hours, while RLVR used 32×40GB A100 GPUs for 56 hours. See Appendix C and Appendix M for full training details and the visualized training dynamics.

# 3 A Comprehensive Evaluation of X-Reasoner

**Evaluation Setup** To assess the generalizability of X-Reasoner, we evaluate its performance across four distinct settings: (1) General-domain text-only tasks; (2) General-domain multimodal tasks (X-modality); (3) Specialized-domain text-only tasks (X-domain); and (4) Specialized-domain multimodal tasks (X-modality&domain) (see Appendix E). The baseline is Qwen-2.5-VL-7B. Inference is conducted using the vLLM backend (Kwon et al., 2023). Consistent with recent reproducibility studies (Hochlehnert et al., 2025), we observe variability even under greedy decoding (temperature=0), due to hardware and environment differences. Therefore, alongside greedy decoding results consistent with prior studies, we report average accuracy, majority-vote accuracy (following Wang et al. (2023)), and pass@n (counting an example correct if any of n attempts is correct) accuracy over five runs at temperature 0.3 to ensure robustness and reproducibility. Unless noted otherwise, all evaluations use CoT prompting (prompt templates are in Appendix D). Additionally, we apply forced-exiting mechanism Section 2.1, capping output generation at 4,096 tokens.

## 3.1 X-Reasoner's Cross-Modality Generalization

In Figure 3, we assess the cross-modality generalization capabilities of X-Reasoner. We first validate its effectiveness on text-only tasks, observing significant performance gains in MMLU-Pro and GSM8K. Moreover, these improvements robustly transfer to multimodal benchmarks, reflected consistently across average, majority vote, and pass@5 accuracy. These results suggest not only superior reasoning ability but also a broader and more effective search space for further improvement.

We further show that X-Reasoner achieves SOTA on MMMU, MMMU-Pro and MathVista, while remaining competitive on others in Table 4. Notably, these SOTA models are extensively trained on multimodal data, underscoring the critical finding that text-only reasoning training alone is sufficient—and likely accounts for the majority of learning necessary for effective multimodal reasoning. In Appendix Q, we present a detailed comparison of text-only and multimodal training, highlighting the limitations of multimodal data alone for inducing long-horizon reasoning and the complementary benefits of combining multimodal and text-only training (Wei et al., 2025).

Qualitative analysis in Appendix N shows that X-Reasoner consistently produces more sophisticated reasoning than the baseline, exhibiting detailed planning, verification (e.g., "wait" steps), and self-correction. In cross-modality tasks, it effectively integrates visual inputs into step-by-step reasoning. For instance, in Table 17, X-Reasoner accurately analyzes all bar values in a plot, avoiding common baseline errors. These examples demonstrate that X-Reasoner internalizes general reasoning patterns through text-only training and successfully transfers them across modalities.

**Takeaway 3.2.1**

X-REASONER, trained on text-only data, consistently improves multimodal task performance, matching or surpassing prior SOTA trained explicitly with multimodal data.

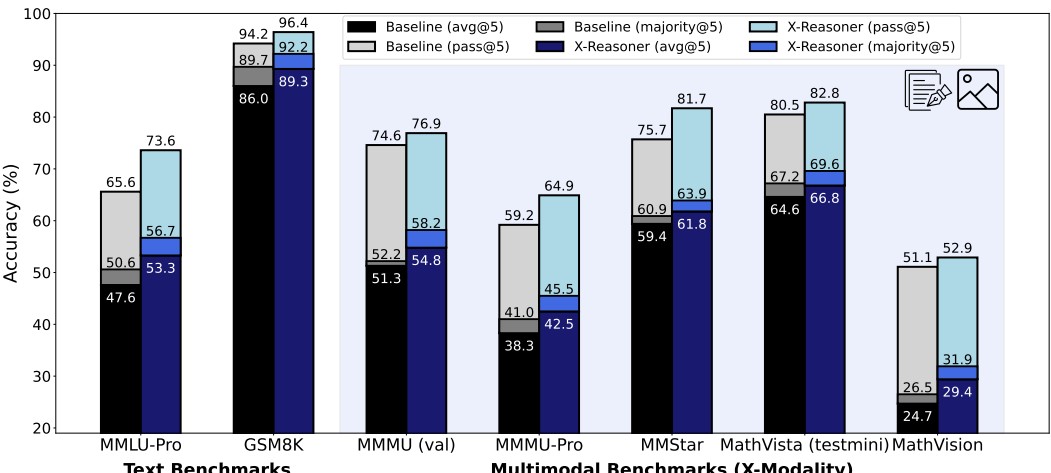

Figure 3: Comparing X-REASONER and baseline on text-only and multi-modal benchmarks (SD reported in Table 13). Despite being trained with text-only data, X-REASONER can significantly improve multi-modal benchmarks, showing the generalization of X-REASONER's reasoning. X-REASONER's consistent gains can be found in additional multimodal benchmarks (Appendix P).

Table 4: X-REASONER outperforms 7B/8B SOTA on multimodal tasks. All results (see Appendix R for notes on replication) are based on greedy decoding. (See Appendix O.2 for additional baselines)

| Model | Reasoning Data Source | MMMU (Val) | MMMU-Pro | MathVista (testmini) | MathVision |
|---|---|---|---|---|---|
| Qwen-VL-2.5-7B-Instruct (Bai et al., 2025) | - | 53.0 | 38.7 | 62.8 | 25.0 |
| R1-Onevision-7B (Yang et al., 2025) | multimodal | - | - | 64.1 | 29.9 |
| MAmmoTH-VL2-7B (Jia et al., 2025) | multimodal | 54.7 | 40.7 | 68.1 | - |
| MM-Eureka-8B (Meng et al., 2025) | multimodal | 49.2 | - | 67.1 | 22.2 |
| Mulberry-7B (Yao et al., 2024) | multimodal | 55.0 | 36.8 | 63.1 | - |
| X-REASONER (7B) | text | **56.4** | **43.0** | **69.0** | 29.6 |

**Ablation Study: Is X-REASONER simply solving the text-solvable examples?** A potential limitation in evaluating vision-language models is their tendency to rely on text-based shortcuts, solving tasks without genuinely integrating visual information. To rigorously confirm that the cross-modality generalization observed with X-REASONER is not merely due to improved text-only shortcut solutions, we conduct an ablation experiment. Specifically, we identify and remove text-solvable examples and assess whether performance gains persist afterward. To identify these text-solvable examples, we mask visual inputs for multimodal tasks previously evaluated in Section 3.1, performing text-only evaluations using both X-REASONER and the baseline `Qwen2.5-VL-7B-Instruct`. For each model, we sample three independent responses per question and eliminate examples consistently solved through text alone. Table 5 summarizes the number of remaining examples after removing text-solvable examples for each task, along with the corresponding model performances. While a substantial number of text-solvable instances were identified, these primarily reflect models' textual reasoning shortcuts rather than dataset issues (Yue et al., 2024b). Crucially, X-REASONER maintains performance advantages over the baseline after text-only solvable examples are excluded, affirming true multimodal reasoning capabilities. This confirms that X-REASONER effectively utilizes visual context within the reasoning framework acquired through text-only training.

**Takeaway 3.2.2**

X-REASONER achieves authentic multimodal reasoning capabilities, as evidenced by sustained improvements after excluding text-only solvable examples.

Table 5: Comparing X-REASONER and baseline performance after removing text-solvable examples. The improvement of X-REASONER remains, indicating true multimodal generalizability.

| Task | #original - #text-solvable = # | Baseline | X-REASONER |
|---|---|---|---|
| MMMU | 900 - 308 = 592 | 39.3 | **41.4** |
| MMMU-Pro | 1730 - 259 = 1471 | 33.4 | **36.4** |
| MathVista (testmini) | 1000 - 257 = 743 | 57.9 | **60.6** |
| MathVision | 3040 - 448 = 2592 | 18.6 | **21.8** |
| MMStar | 1500 - 259 = 1241 | 56.6 | **59.0** |

## 3.2 X-REASONER IN MEDICINE: A CROSS-MODALITY AND CROSS-DOMAIN STUDY

In this section, we examine the cross-domain and cross-modality transfer capabilities of X-REASONER, specifically within medical contexts. While X-REASONER is trained exclusively on general-domain text data, we further probe the benefits of domain-specific adaptation by developing X-REASONER-MED, which extends X-REASONER with additional training using medical text data.

**Evaluation Setup** Our evaluation includes three text-only medical tasks: MMLU-Pro-Health (the healthcare subset of MMLU-Pro (Wang et al., 2024b) following Chen et al. (2024a)), MedQA (the original USMLE version), and MedQA (4-ops: the four-choice version). For multimodal medical evaluations, we utilize healthcare subsets of MMMU (Yue et al., 2024a) and MMMU-Pro (Yue et al., 2024b), designated as MMMU-Health and MMMU-Pro-Health respectively. We also include MedXpertQA-MM (Zuo et al., 2025), OmniMedVQA (Hu et al., 2024) and NEJM Image Challenge (New England Journal of Medicine, 2025), which focus on medical knowledge and reasoning. We additionally test X-REASONER's generalization beyond medicine in Appendix O.1.

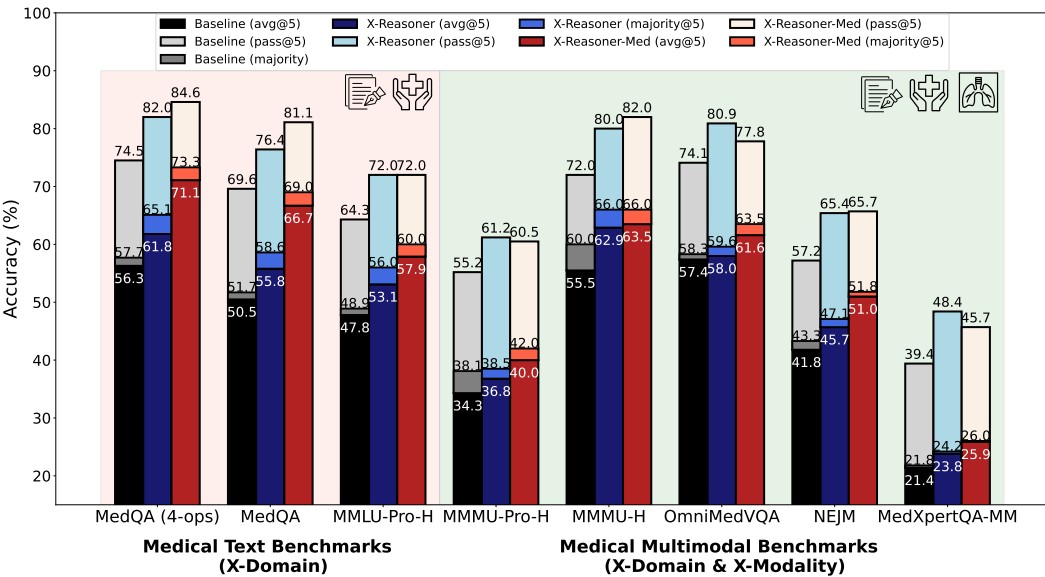

Figure 4: Comparing X-REASONER, X-REASONER-MED and baseline on text and multimodal medical benchmarks (SD. reported in Table 14). X-REASONER, trained with general-domain text-only data brings consistent improvement across medical tasks. X-REASONER-MED, obtained by continued training of X-REASONER on medical text data, further improves performance.

**Results** Figure 4 shows X-REASONER demonstrates robust cross-domain and cross-modality generalization capabilities, surpassing baseline models on both textual and multimodal medical tasks. Remarkably, as detailed in Figure 1, X-REASONER surpassing or matching previous SOTA on MedXpertQA-MM and MMMU-Health, despite previous SOTA models being explicitly trained on multimodal medical data. Qualitative assessments in Appendix N further illustrate X-REASONER's effective integration of visual information and medical domain-specific reasoning. As an example, X-REASONER accurately identifies medical patterns, such as correctly recognizing white matter abnormalities indicative of progressive multifocal leukoencephalopathy in histology slides (Table 18).

> **Takeaway 3.3.1**
>
> X-REASONER substantially enhances medical task performance, confirming its strong capability for cross-domain and cross-modality generalization.

**X-REASONER-MED: Enhancing Medical Performance** We investigate whether additional medical-domain text data can further enhance X-REASONER's medical performance. To explore this, we extend X-REASONER's training with text-only MedQA data. Specifically, starting from X-REASONER, we perform SFT with distilled CoTs from QwQ-32B (Qwen Team, 2025), followed by RLVR on the same data (For ablation, we also compared MedQA SFT + RL with X-REASONER-MED in Appendix K, showing X-REASONER-MED is better, highlighting the benefit of starting domain adaptation from a generalist X-REASONER).

The resulting model, X-REASONER-MED, achieves further improvements across all medical benchmarks. As depicted in Figure 4, X-REASONER-MED consistently surpasses X-REASONER in terms of average and majority vote accuracies. Notably, X-REASONER sometimes attains higher pass@n scores, suggesting it explores a broader search space. Conversely, X-REASONER-MED, benefiting from targeted medical-domain fine-tuning, already leverages this search space more effectively but potentially with reduced room for further gains. Nevertheless, as highlighted in Figure 1, X-REASONER-MED sets new SOTA performance for both text-only and multimodal medical tasks, validating our hypothesis that combining general text-based reasoning with domain-specific text-based fine-tuning unlocks substantial additional performance gains for specialized domains.

> **Takeaway 3.3.2**
>
> X-REASONER-MED, obtained by further training X-REASONER on medical text, improves further and sets new SOTA on both text and multimodal medical benchmarks.

## 4 RELATED WORK

In RL-based learning of multimodal reasoning, most methods such as VFT (Liu et al., 2025b), MM-Eureka (Meng et al., 2025), Vision-R1 (Huang et al., 2025a), R1-V (Chen et al., 2025), and Zhou et al. (2025); Deng et al. (2025); Wan et al. (2025) refine reasoning in VLM with multi-modal training data. Peng et al. (2025b) present a two-stage RL approach that leverages both text-based and multimodal data. On the SFT side, reasoning is commonly distilled from captions or responses synthesized by VLMs. Methods such as MaMMOTH (Guo et al., 2024; Jia et al., 2025), Vision-R1 (Huang et al., 2025a), R1-OneVision (Yang et al., 2025), and Llava-CoT (Xu et al., 2024) follow this strategy to infuse reasoning into multimodal models during SFT. Alternatively, some methods incorporate visual grounding by coupling a text-based reasoning model with a vision encoder (Peng et al., 2025a). Closest to our study, Du et al. (2025) also explores text-only training for multimodal tasks, though their setting is limited to SFT and general-domain evaluation. Recently, there has also been growing interest in applying reasoning techniques to specialized domains such the medical domain, initially focusing on the text modality (Zhang et al., 2025a; Huang et al., 2025b; Chen et al., 2024a). Subsequently, researchers have begun exploring RL-based training on multimodal medical datasets to enhance multimodal reasoning performance in the medical domain, e.g., Pan et al. (2025) and Lai et al. (2025).

## 5 CONCLUSION

This study demonstrates that general domain text-only SFT+RL is the most effective recipe for learning generalizable reasoning even for multi-modal and out-of-domain settings. Building on this insight, we introduce X-REASONER, a simple yet effective post-training recipe that enhances the reasoning capabilities of VLMs using only general-domain text-based supervision. Through extensive experiments, we show that X-REASONER generalizes effectively across modalities and domains, surpassing prior SOTA trained with in-domain multimodal data. We further show X-REASONER serves as a strong foundation for domain specialization by introducing X-REASONER-MED, a variant further trained on medical text, which achieves new SOTA on various medical benchmarks across both text-only and multimodal settings. We leave to future work the exploration of continued in-domain multimodal SFT/RL and we provide detailed discussion on our limitations in Appendix A.

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

## A  LIMITATIONS

While we show X-REASONER effectively equips vision-language models with generalizable reasoning capabilities, we acknowledge several limitations that offer opportunities for future work:

**Base Model Constraints** Prior studies have emphasized the importance of starting from pretrained base checkpoints, rather than instruction-tuned models, to better incentivize the emergence of reasoning behaviors (DeepSeek-AI et al., 2025; Hu et al., 2025). However, in our case, we are constrained by computational resources and the lack of base VLMs in the recent open-source releases, limiting our ability to empirically validate these claims.

**Model Scale and Backbone Diversity** Prior work suggests that larger models tend to exhibit stronger reasoning capabilities (DeepSeek-AI et al., 2025). However, due to computational constraints, our model size is limited to 7B parameters. We also focused exclusively on the Qwen-VL series and did not evaluate other publicly available VLMs.

**Task Scope** Our evaluation primarily targets mathematical questions and multiple-choice questions across general and medical domains. While these benchmarks provide clear, verifiable supervision for assessing reasoning capabilities, they do not capture the full spectrum of real-world reasoning challenges. In particular, we have not tested our approach on open-ended generation, interactive dialogue, or instruction-following scenarios. Understanding how reasoning generalizes to such unconstrained settings remains an important area for future exploration. Our evaluation settings also focus on two modalities—text and vision—which are currently the most central to vision-language foundation models. Extending this framework to additional modalities such as audio and video remains an important and valuable direction for future work

## B  GRPO

Let $\pi_\theta$ be a vision language model (VLM) based on decoder-only transformers and parameterized by $\theta$. It takes as input a sequence of tokens, denoted by $\mathbf{q}$, such as a question, and autoregressively decodes the response, denoted by $\mathbf{o}$. Optionally, $\pi_\theta$ can also take visual features, such as pictures, as input that is tokenized by a vision encoder. However, visual features are not included in our training paradigm, and $\mathbf{q}$ during training is supposed to be natural languages only. We use text-only math datasets for training. Let $\mathcal{D}$ be a dataset and $(\mathbf{q}, \mathbf{a}) \in \mathcal{D}$ be pairs of questions and answers.

In Group Relative Policy Optimization (Shao et al., 2024, GRPO), for a given query $\mathbf{q}$, the policy model samples a group of responses, and the token-level advantage $\hat{A}_{i,t}$ for the $i$-th response is estimated by normalizing its reward $r_i$ relative to the group rewards $\mathbf{r}$:

$$\hat{A}_{i,t} = \frac{r_i - \mathrm{mean}(\mathbf{r})}{\mathrm{std}(\mathbf{r})}, \tag{1}$$

Similar to PPO (Schulman et al., 2017), GRPO employs a clipped surrogate objective but includes a KL-divergence penalty term directly in its loss function to encourage stability:

$$\mathcal{J}_{\mathrm{GRPO}}(\theta) = \mathbb{E}_{\mathbf{q} \sim \mathcal{Q}, \{\mathbf{o}_i\}_{i=1}^G \sim \pi_{\theta_{\mathrm{old}}}(\cdot|\mathbf{q})} \left\{ \frac{1}{G} \sum_{i=1}^G \frac{1}{|\mathbf{o}_i|} \sum_{t=1}^{|\mathbf{o}_i|} \min \left[ \frac{\pi_\theta(o_{i,t} \mid \mathbf{q}, \mathbf{o}_{i,<t})}{\pi_{\theta_{\mathrm{old}}}(o_{i,t} \mid \mathbf{q}, \mathbf{o}_{i,<t})} \hat{A}_{i,t}, \right. \right.$$

$$\left. \left. \mathrm{clip} \left( \frac{\pi_\theta(o_{i,t} \mid \mathbf{q}, \mathbf{o}_{i,<t})}{\pi_{\theta_{\mathrm{old}}}(o_{i,t} \mid \mathbf{q}, \mathbf{o}_{i,<t})} \hat{A}_{i,t}, 1 - \epsilon, 1 + \epsilon \right) \right] - \beta \mathbb{D}_{\mathrm{KL}} \left[ \pi_\theta \| \pi_{\mathrm{ref}} \right] \right\}, \tag{2}$$

where $\pi_\theta$ is the policy model, i.e., VLM. For a query $\mathbf{q}$, we sample $G$ outputs $\{\mathbf{o}_1, \ldots, \mathbf{o}_G\}$ from the old policy model $\pi_{\theta_{\mathrm{old}}}$. Clip ratio $\epsilon$ and $\beta$ are hyper-parameters and $\pi_{\mathrm{ref}}$ is the reference VLM.

## C  HYPERPARAMETERS

In this section, we discuss the hyperparameters used in our SFT and RL experiments.

## C.1 SUPERVISED FINE-TUNING

For the SFT experiments, we use the AdamW optimizer (Loshchilov & Hutter, 2019) with a learning rate of $1 \times 10^{-5}$, weight decay of 0.0, $\beta_1 = 0.9$ and $\beta_2 = 0.999$. We adopt a multiplicative learning rate scheduler with a decay rate of 0.8 for each epoch. The training batch size is set as 4 per device and we used 32 GPUs in total. For training efficiency, we remove examples that exceed 4096 tokens and result in around 40k training examples. Each dataset will be trained for 4 epochs.

## C.2 REINFORCEMENT LEARNING

Throughout our experiments, we set KL coefficient $\beta = 1.0 \times 10^{-2}$ and the clip ratio $\epsilon = 0.2$. For each example, we sample 8 responses with a maximum length of 4096 tokens and sampling temperature of 1.0.

When updating the actor model, we use AdamW (Loshchilov & Hutter, 2019) and set the learning rate as $3.0 \times 10^{-6}$ and weight decay as $1.0 \times 10^{-2}$. A warm-up learning rate scheduler is used by using 10% of the total training steps. Gradient norm is applied with a threshold of 1.0. A global batch size (calculated across all devices) is set as 128. All the training are stopped on the convergence of the reward value on the training set.

## D PROMPT TEMPLATES

Below are the prompt templates in Jinja format Pallets (2024) used in training and evaluation. Both Qwen-2.5-VL-7B-Instruct baseline and X-REASONER are able to successfully follow instructions to enclose answers within `<answer> </answer>` tags with a success rate of around 99% . Therefore the improvements from X-REASONER are not attributable to formatting.

---

**Prompt Template for Training**

```
You will solve a problem/request. You should provide your
thoughts within <think> </think> tags before
providing the answer.\nWrite your
final answer within <answer> </answer> tags.
Here is the question:
{{ question }}{% if options %}\nOptions:
\n\n{{ options }}{% endif
%}\n\n
```

---

**Prompt Template for Evaluation Multi-choice Questions**

```
You should provide your thoughts within <think> </think> tags,
then answer with just one of the options below within
<answer> </answer> tags (For example,
if the question is \n'Is the earth flat?\n A: Yes
\nB: No', you should answer with <think>...</think>
<answer>B: No</answer>).
Here is the question: {{ question }}{% if options %}\n
Options:\n\n{{ options }}{% endif %}\n\n
```

```
Prompt Template for Evaluation Mathematical Questions

A conversation between User and Assistant. The user asks a
question, and you as the assistant solves it.
You should first think about the reasoning process
in the mind and then provide the user with the answer.
The reasoning process and answer are enclosed within
<think> </think> and <answer> </answer> tags, respectively,
i.e., <think> reasoning process here </think>
<answer> the final answer as the option letter
or the number depending on the question </answer>
(For example, if the question is \n'Is the earth flat?\n
A: Yes \nB: No', you should answer with <think> your reasoning
</think> <answer>B: No</answer>. If the question is
'What is 1+1?', you should answer with <think>
your reasoning </think> <answer>2
</answer>).\n\nHere is the question: {{ question }}
{% if options %}\nOptions:\n\n{{ options }}{% endif %}\n\n
```

Table 6: Tasks for different evaluation settings. CC: CC BY-SA 4.0

| Evaluation setting | Task name [license] | Shorthand | Data size |
|---|---|---|---|
| General-domain text-only | GSM8K main [MIT] (Cobbe et al., 2021) | GSM8K | 1,319 |
| | MMLU-Pro [MIT] (Wang et al., 2024b) | MMLU-Pro | 12,032 |
| General-domain multimodal (X-modality) | MMMU (val) [CC] | MMMU | 900 |
| | MMMU-Pro [Apache-2.0] (Yue et al., 2024b) | MMMU-Pro | 1,730 |
| | MMStar [CC] (Chen et al., 2024b) | MMStar | 1,500 |
| | MathVista (testmini) [CC] (Lu et al., 2024) | MathVista | 1,000 |
| | MathVision [MIT] (Wang et al., 2024a) | MathVision | 3,040 |
| Specialized-domain text-only (X-domain) | MedQA [MIT] (Jin et al., 2021) | MedQA | 1273 |
| | MedQA (4 options) [MIT] (Jin et al., 2021) | MedQA (4-ops) | 1,273 |
| | MMLU-Pro-Health [MIT] (Wang et al., 2024b) | MMLU-Pro-H | 818 |
| Specialized-domain multimodal (X-modality & X-domain) | MMMU-Health (Yue et al., 2024a) | MMMU-H | 150 |
| | MMMU-Pro-Health [CC] (Yue et al., 2024b) | MMMU-Pro-H | 286 |
| | MedXpertQA-MM [MIT] (Chen et al., 2024b) | MXQ | 2,000 |
| | OmniMedVQA [CC Zero] (Hu et al., 2024) | OMV | 1,000 |
| | NEJM Image Challenge (New England Journal of Medicine, 2025) | NEJM | 947 |

## E  EVALUATION TASKS

Table 6 summarizes tasks used for different evaluation settings in our experiments: general-domain text-only, general-domain multimodal, specialized-domain text-only, and specialized-domain multimodal. We focus specifically on the medical domain as our representative specialized domain due to its significant practical importance and recent intensive research activities (Li et al., 2023; Zambrano Chaves et al., 2025; Zhang et al., 2025b; Luo et al., 2022; Nori et al., 2023; 2024; Codella et al., 2024; Poon et al., 2023; Xie et al., 2023; Ness et al., 2024). For OmniMedVQA, we sample 1000 from the full data for efficient evaluations.

## F  PREVIOUS SOTA RESULTS

Table 7 lists details of the previous SOTA results for each task.

Table 7: Previous SOTA results. * indicates our replicated results.

| Task | Previous SOTA Model | Result |
|---|---|---|
| GSM8K | MMOS-DeepSeekMath-7B (Chen et al., 2024c) | 87.2 |
| MMLU-Pro | Gemma-2-9B-it (Gemma Team et al., 2024) | 52.1 |
| MMMU-Pro | MAmmoTH-VL2-7B (Jia et al., 2025) | 40.7 |
| MMMU (Val) | Mulberry-7B (Yao et al., 2024) | 55.0 |
| MedQA (4-ops) | HuatuoGPT-o1-7b (Chen et al., 2024a) | 71.6* |
| MMLU-Pro-H | HuatuoGPT-o1-7B (Chen et al., 2024a) | 54.3* |
| MMMU-H | GMAI-VL-RL (Su et al., 2025) | 57.3 |
| MedXpertQA-MM | GMAI-VL-Rl (Su et al., 2025) | 23.8 |

## G  Cross-distribution/task Generalization of SFT

We conduct a cross-distribution investigation where we fine-tune an instruction-tuned VLM, `Qwen2.5-VL-7B-Instruct` (Bai et al., 2025) using text-only MedQA (Jin et al., 2021), a specialized medical QA dataset. The training signal comprises detailed long-COT reasoning traces, distilled via rejection sampling (Huang et al., 2023; Zelikman et al., 2022) from QwQ-32B (Qwen Team, 2025). We then evaluate the model's generalization on an out-of-distribution task: MMLU-Pro-Health, the healthcare subset of MMLU-Pro (Wang et al., 2024b). The full training and evaluation details are provided in Sections 2.3 and 3. As shown in Table 8, all the SFT approaches consistently transfers the improvements from the source task (MedQA) to the target task (MMLU-Pro-Health), demonstrating strong cross-task generalization from text-only SFT.

Table 8: Comparing baseline and text-only SFT with and without CoT On MedQA task and evaluating on in & out of distribution tasks

| Training: MedQA Eval: in & out of distribution | Modality | Non-CoT | | CoT | |
|---|---|---|---|---|---|
| | | Baseline | SFT | Baseline | SFT |
| MedQA (In distribution) | Textual | 49.3 | 57.6 (+8.3) | 50.5 | **61.9 (+11.4)** |
| MMLU-Pro-H (Out of dist.) | Textual | 43.7 | 47.2 (+3.5) | 47.8 | **54.2  (+6.4)** |

## H  Forced-Exiting to mitigate endless thinking in long-CoT SFT

While long-CoT SFT effectively learns transferrable reasoning patterns, such as self-reflection, verification, and correction (see qualitative analyses in Appendix N), we observe a recurring challenge: the model occasionally fails to terminate its responses. Specifically, the model continues generating tokens indefinitely without producing a stop signal. Such endless thinking occurs 13% of generations for MMLU-Pro-Health tasks and 32% for MMMU-Pro, as reported in (Table 9). Upon closer inspection, these failure cases are often characterized by verbose, repetitive output, where the model persistently attempts to "re-think" or "double-check", an artifact likely stemming from the reasoning mechanism introduced during long-CoT SFT (DeepSeek-AI et al., 2025).

To address this, we implement a forced-exiting mechanism inspired by the method introduced in Muennighoff et al. (2025). This involves appending a designated stop token '`</think>`' once the output sequence reaches a predefined length threshold. This soft intervention encourages the model to wrap up its reasoning process within a reasonable token budget. As shown in Table 9, applying forced-exiting effectively reduces endless thinking, thereby improving final task accuracy.

## I  Comparison of RL and SFT on MedQA data

We conduct a direct comparison between Rl, SFT and their combination with the same MedQA data. As shown in Table 10, pure RL performs worse than SFT and the hybrid SFT + RL performs the best.

Table 9: Forced-exiting mitigates endless thinking and improves final task accuracy from CoT SFT

| Task | Before Forced-Exiting | | After Forced-Exiting | |
|---|---|---|---|---|
| | % endless thinking | Accuracy | % endless thinking | Accuracy |
| MMMU-Pro | 32.5% | 32.5 | 1.4% | 40.4 |
| MMLU-Pro-Health | 13.4% | 46 | 0.1% | 50.4 |

Table 10: Comparison of performance from RL, SFT, and SFT + RL in both in-distribution (ind. text) and generalization settings, including out-of-distribution (ood. text) and cross-modality (X-modality). All models are trained on MedQA data.

| Method | MedQA (ind. text) | MMLU-Pro-Health (ood. text) | MMMU-Pro-Health (X-modality) | Average Response Length (number of words) |
|---|---|---|---|---|
| baseline | 50.5 | 47.8 | 34.3 | 111 |
| SFT | 61.9 | 54.2 | 39.2 | 950 |
| RL | 57.8 | 53.6 | 35.8 | 103 |
| SFT + RL | **64.9** | **55.8** | **39.7** | 1039 |

## J  COMPARING SFT ON OPENTHOUGHTS AND ORZ MATH DATA

As in Table 11, we also experimented with using the Orz math data as well in the SFT stage by distilling CoT traces from QwQ-32B. We found that the X-Reasoner recipe with OpenThoughts SFT performs better than Orz math SFT. This suggests that we need more diverse general-domain training data than math in the SFT stage.

Table 11: Comparing SFT on OpenThoughts and Orz math data in the pipeline

| Method | MMMU-Pro | MathVision |
|---|---|---|
| X-Reasoner (OpenThoughts SFT + Math RL) | 42.5 | 29.4 |
| Math SFT + Math RL | 40.5 | 26.7 |

## K  COMPARING MEDQA SFT + RL WITH X-REASONER-MED

As in Table 12, we also compare MedQA SFT + RL with X-REASONER-MED, which fine-tunes the general-domain X-REASONER using domain-specific data. X-REASONER-Med yields the strongest results, highlighting the benefit of starting from a generalist model and then applying domain-specific fine-tuning.

## L  STANDARD DEVIATION

We report standard deviations for each experiment from the cross-modal evaluation (Figure 3) and cross-domain evaluation (Figure 4) in Table 13 and Table 14. We also report standard deviation for the RL training data experiments from Table 2 in Table 15.

## M  TRAINING DYNAMICS

Figure 5 shows the RL training dynamics when X-REASONER is being trained on Orz (Hu et al., 2025) and when X-REASONER-MED is being trained on MedQA (Jin et al., 2021), including training reward, policy gradient loss, mean advantage, mean training response length, training response clip ratio, validation accuracy.

Table 12: Comparing MedQA SFT + RL and X-REASONER-MED

| Method | MedQA | MMLU-Pro-H | MMMU-Pro-H | MedXpertQA-MM |
|---|---|---|---|---|
| MedQA SFT + RL | 64.9 | 55.8 | 39.7 | 24.4 |
| X-REASONER-MED | 66.7 | 57.9 | 40.0 | 25.9 |

Table 13: Standard Deviation for the Cross-Modality Evaluation in Figure 3 comparing X-REASONER and the baseline across 5 runs.

| Method | GSM8k | MMLU-Pro | MMMU | MMMU-Pro | MMStar | MathVista (testmini) | MathVision |
|---|---|---|---|---|---|---|---|
| Baseline | 0.22 | 0.29 | 0.61 | 0.39 | 0.47 | 0.10 | 0.59 |
| X-REASONER | 0.32 | 0.24 | 1.05 | 0.90 | 0.65 | 0.77 | 0.38 |

Table 14: Standard Deviation for evaluating X-REASONER, X-REASONER-MED and the baseline across 5 runs on medical benchmarks from Figure 4. MedQA (4): MedQA (4-ops), OMV: Omni-iMedVQA, MXQ: MedXpertQA-MM

| Method | MedQA (4) | MedQA | MMLU-P-H | MMMU-P-H | MMMU-H | OMV | NEJM | MXQ |
|---|---|---|---|---|---|---|---|---|
| Baseline | 0.80 | 0.38 | 0.72 | 1.72 | 3.7 | 0.65 | 0.45 | 0.29 |
| X-REASONER | 0.89 | 0.87 | 0.61 | 1.49 | 2.71 | 0.96 | 0.81 | 0.48 |
| X-REASONER-MED | 0.85 | 0.46 | 0.78 | 0.65 | 4.76 | 0.96 | 0.42 | 0.88 |

Table 15: Standard Deviation for the results from the Rl training data exploration in (Table 2)

| Method | GSM8K | MMMU-Pro | MMLU-Pro-Health | MMMU-Pro-Health |
|---|---|---|---|---|
| Baseline | 0.22 | 0.39 | 0.72 | 1.72 |
| Math orz RL | 0.49 | 0.78 | 1.13 | 2.10 |
| MedQA RL | 0.50 | 0.80 | 0.86 | 1.7 |
| ThinkLite RL | 0.51 | 0.30 | 0.76 | 1.24 |
| OmniMedVQA RL | 0.24 | 0.40 | 0.69 | 0.92 |

# N    QUALITATIVE ANALYSIS

Table 16 shows some example reasoning output from X-REASONER as compared with the baseline. We can clearly see that X-REASONER is able to engage in more complex reasoning traces including detailed planning, verification ("wait). This complex reasoning patterns is successfully transferred to cross-modality setups where X-REASONER seamlessly integrates the vision information into the reasoning chain, and provides detailed step-by-step reasoning with verification for review/correction. In the example, X-REASONER provides a detailed plan to go over all the bar values whereas the baseline misses half of the bar values. X-REASONER is also able to correct its mistake in the reasoning chain. In the cross-modal cross-modality setting, X-REASONER connects the dots from both visual information and domain knowledge to the reasoning chain, and successfully identifies the patterns (white matter) from the histology slide and refers to its medical domain knowledge to connect it to progressive multifocal leukoencephalopathy.

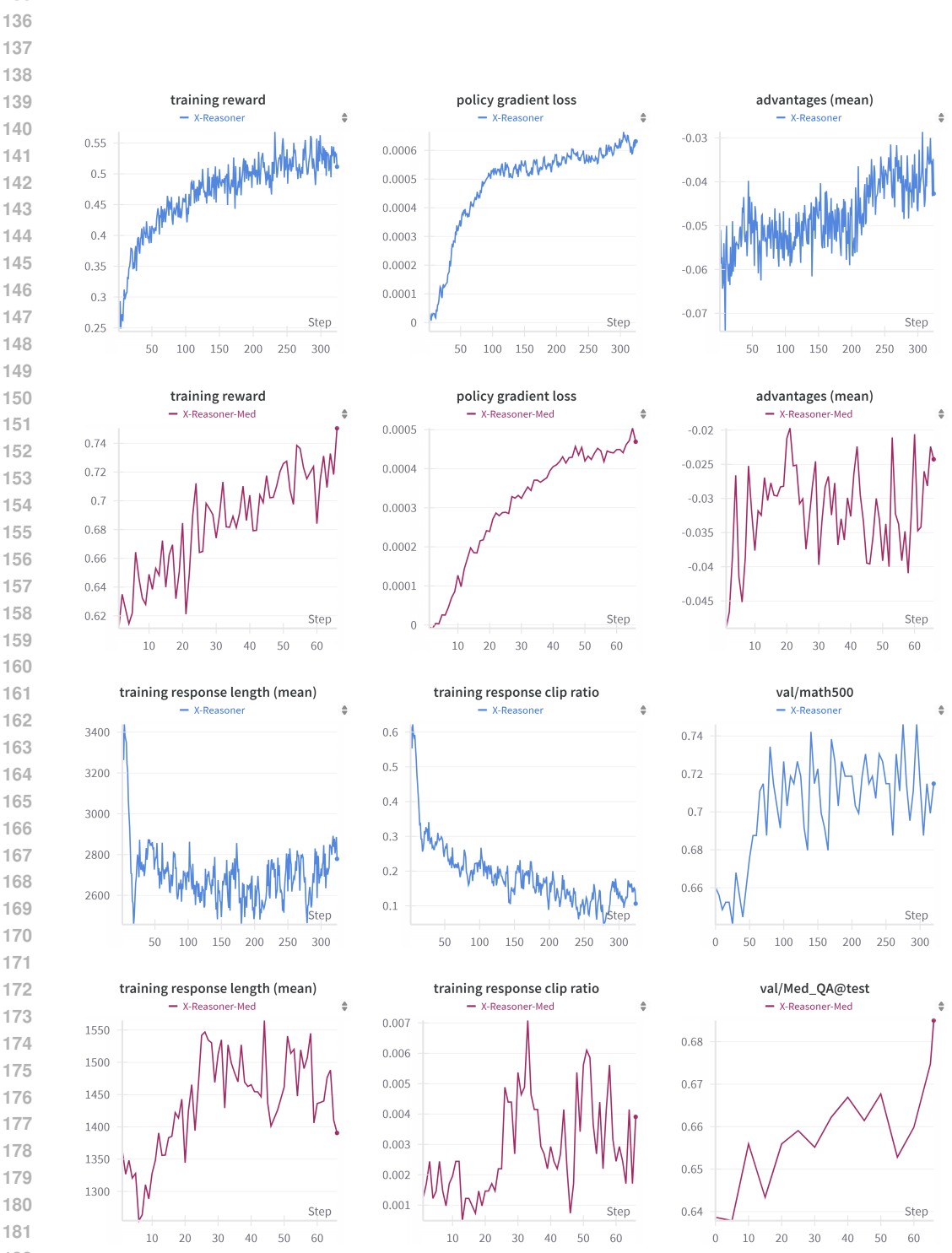

Figure 5: RL training dynamics of X-REASONER and X-REASONER-MED

Table 16: Reasoning output comparison between the baseline `Qwen2.5-VL-7B-Instruct` model and X-REASONER on an text-only question.

| Question | The great dragon, Perg, sat ...within a distance of 1000 feet. Polly could throw the gold javelin...for a distance of 400 feet, well within the reach of the dragon's flames. But when Polly held the sapphire gemstone, she could throw the javelin three times farther than when not holding the gemstone. If holding the gemstone, how far outside of the reach of the dragon's flames could Polly stand and still hit the dragon with the gold javelin? (from gsm8k main) |
|---|---|
| **Response** | |
| **Baseline** | `<think>` To solve this problem, we need to determine the maximum distance Polly can stand from the dragon and still hit it with the javelin when holding the sapphire gemstone. Here's the step-by-step reasoning: 1. When Polly is not holding the gemstone, she can throw the javelin for a distance of 400 feet. 2. When Polly holds the gemstone, she can throw the javelin three times farther than when not holding the gemstone. Therefore, the distance she can throw the javelin when holding the gemstone is $\( 400 \times 3 = 1200 \)$ feet...`</think>` `<answer>`1200`</answer>` ✗ |
| **X-REASONER** | `<think>`Okay, let's see. **First, let's break down the information.** Without the gemstone, the javelin's distance is 400 feet. With the gemstone, it's three times that. So, 400 feet multiplied by 3 gives the new throwing distance. Let me calculate that: 400 * 3 = 1200 feet... **Wait**, but is that correct? Let me check again. Wait, the problem says "how far outside of the reach of the dragon's flames could Polly stand and still hit the dragon with the gold javelin?" So, if the dragon's flames reach 1000 feet, then the maximum distance Polly can stand from the dragon's position and still hit it is 1200 feet. But the question is asking how far outside the dragon's flames she can stand. So, 1200 feet minus 1000 feet is 200 feet. Therefore, the answer should be 200 feet. `</think>` `<answer>`200`</answer>` ✓ |

Table 17: Reasoning output comparison between the baseline `Qwen2.5-VL-7B-Instruct` model and X-REASONER on a multimodal question.

| **Question** | How many bars have value below 40? (from MathVista) |
| --- | --- |

| | **Response** |
| --- | --- |
| **Baseline** | `<think>`To determine how many bars have a value below 40, I'll examine each bar's value: 

 1. 2014: 63 (above 40) 
 2. 2015: 95.5 (above 40) 
 3. 2016: 82.2 (above 40) 
 4. 2017: 72.9 (above 40) 
 5. 2018: 58 (above 40) 
 6. 2019: 55.2 (above 40) 
 None of the bars have a value below 40.`</think><answer>0</answer>` ✗ |
| **X-REASONER** | `<think>` Okay, let's see. The question is asking how many bars have a value below 40. The image is a bar chart with two categories: Banknotes and Coins, and years from 2014 to 2019. Each year has two bars, one for each category. The values are given in thousands, so we need to check each bar's value and count those below 40 thousand. 
 **First, let's go through each year**: 
 2014: 
 - Banknotes: 63 thousand. That's above 40. 
 - Coins: 46 thousand. Also above 40. 

 ... 
 Now, let's count the bars below 40: 
 - 2016: Coins (33) 
 - 2017: Coins (32.5) 
 - 2018: Coins (33.1) 
 - 2019: Coins (42.1) - **Wait**, 42.1 is above 40. Wait, hold on. Wait, 2019's Coins is 42.1 thousand. 42.1 is above 40. So that's not below 40. So only the 2016 and 2017 Coins are below 40... Let's recast: 

 2014: Banknotes 63 (no), Coins 46 (no) → 0 
 2015: Banknotes 95.5 (no), Coins 46 (no) → 0 
 2016: Banknotes 82.2 (no), Coins 33 (yes) → 1 
 2017: Banknotes 72.9 (no), Coins 32.5 (yes) → 1 
 2018: Banknotes 58 (no), Coins 33.1 (yes) → 1 
 2019: Banknotes 55.2 (no), Coins 42.1 (no) → 0 
 ... So total bars below 40: 3. Therefore, the answer is 3. 
 `</think>` 
 `<answer>3</answer>` ✓ |

Table 18: Reasoning output comparison between the baseline `Qwen2.5-VL-7B-Instruct` model and X-REASONER on an medical multimodal question.

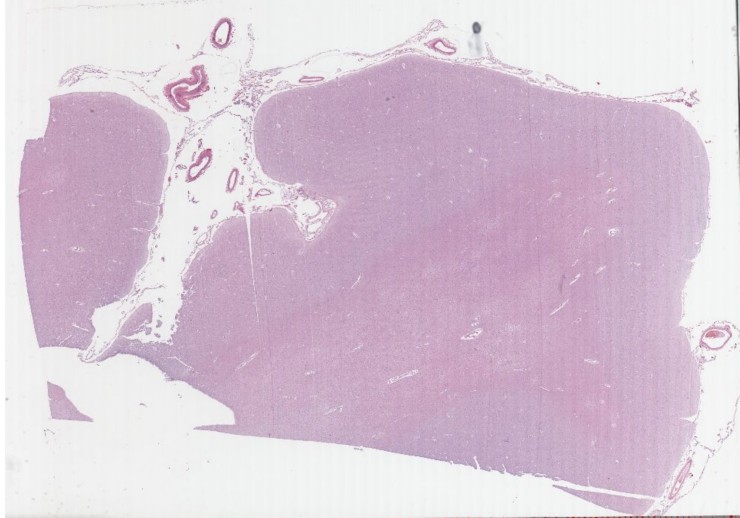

| Question | What is in the image? A. Progressive Multifocal Leukoencephalopathy (PML) B. Rabies C. West Nile Virus D. Herpes Simplex encephalitis E. Encephalitis Lethargica F. HIV encephalitis G. Lyme Disease H. Syphilis I. Meningitis J. Creutzfeldt-Jakob disease (from MMMU_PRO HEALTH) |
|---|---|
| | **Response** |
| **Baseline** | `<think>`The image appears to be a histological section, likely from a brain biopsy, given the presence of cellular structures and the context of the question. The pattern of the lesions and the cellular infiltration are characteristic of encephalitis. Herpes Simplex encephalitis (HSE) is a well-known cause of encephalitis, often presenting with a characteristic pattern of perivascular inflammation and necrosis, which can be seen in such images. The lesions in the image seem to have a focal nature, which is consistent with HSE. However, without further clinical information and a more detailed examination, a definitive diagnosis cannot be made. Given the options, the most likely answer is: D. Herpes Simplex encephalitis`</think><answer>`D. Herpes Simplex encephalitis`</answer>` ✗ |
| **X-REASONER** | `<think>` Okay, let's try to figure out this question. The image shows a histological section with some lesions. The question is about which neurological disease is depicted. **Let me go through the options one by one.** First, I remember that PML (Progressive Multifocal Leukoencephalopathy) is caused by the JC virus and typically affects the white matter of the brain, leading to lesions that can be seen on MRI as areas of low signal intensity on T2-weighted images. The histology might show demyelination and gliosis. The image here shows some irregular areas with a pinkish hue, which could be consistent with demyelination. **But I need to check** if the lesions are in the white matter. The image doesn't show the entire brain, but the lesions appear to be in the white matter, which is typical for PML. Rabies usually presents with encephalitis, but the histology isn't typically as specific as PML... Encephalitis Lethargica is an old disease, but its histology isn't as specific as PML. Given the appearance of the lesions in the white matter, the pinkish areas, and the possibility of demyelination, PML seems the most likely. So the answer is A. `</think>` `<answer>A</answer>` ✓ |

## O    EXTENSION OF X-REASONER TO DIFFERENT MODEL SIZES AND DIFFERENT MODEL FAMILIES

We provide below the generalization evaluation of the X-Reasoner recipe on a different model size (Qwen-2.5-VL-3B (Bai et al., 2025)) and on a different model family (GLM-4.1V-9B (Team et al., 2025)). We observe that the trend remains consistent across both model sizes and model families, where the text-only X-Reasoner recipe consistently improves the performance of the base model on all multimodal and domain-specific tasks.

Table 19: Generalization evaluation of X-Reasoner across different model sizes and model families.

| Model | MMMU | MMMU-Pro | MathVision(testmini) | MMMU-Pro-Health | MMMU-Health |
|---|---|---|---|---|---|
| Qwen-2.5-3B-VL | 44.8 | 31.5 | 18.1 | 28.6 | 50.0 |
| + X-REASONER | **51.0** | **35.8** | **30.3** | **30.1** | **52.6** |
| GLM-4.1V-9B | 53.0 | 43.7 | 32.6 | 41.9 | 61.3 |
| + X-REASONER | **60.2** | **48.2** | **35.5** | **48.9** | **62.0** |

### O.1    X-REASONER'S CROSS-DOMAIN GENERALIZATION

Apart from the medical domain, we further evaluate X-Reasoner on two additional domains: the history domain from MMLU-History and the finance domain from MMMU-Pro-Finance, and report the results below. We show that X-Reasoner's cross-domain generalization extends beyond the medical domain to other specialized domains, as demonstrated by the substantial gains on both the History and Finance benchmarks.

Table 20: Cross-domain evaluation on History and Finance benchmarks.

| Model | History domain | Finance domain |
|---|---|---|
| Qwen-2.5-VL-7B | 42.7 | 45.3 |
| X-Reasoner | **44.6** | **50.0** |

### O.2    COMPARING X-REASONER WITH MORE RECENT BASELINES

Below we compare X-REASONER with more recent baseline models that are also built on top of Qwen-2.5-VL-7B. We observe that X-REASONER performs competitively with these stronger multimodal reasoning baselines, and in fact still achieves state-of-the-art performance across both general-domain and medical-domain multimodal benchmarks.

Table 21: Comparison with recent Qwen-2.5-VL-7B–based multimodal reasoning models.

| Model | MMMU | MMMU-Pro | MMMU-H | NEJM |
|---|---|---|---|---|
| Qwen baseline | 53.0 | 38.7 | 55.5 | 41.8 |
| X-REASONER | **56.4** | **43.0** | **62.9** | **45.7** |
| VL-Rethinker-7B (Wang et al., 2025a) | 56.3 | 41.7 | 60.7 | 43.5 |
| ThinkLite-VL-7B (Wang et al., 2025b) | 55.5 | 39.0 | 62.0 | 45.0 |

## P   EVALUATING X-REASONER ON ADDITIONAL MULTIMODAL BENCHMARKS

Below we report X-REASONER 's performance on additional multimodal benchmarks. X-REASONER consistently surpasses the baseline model, aligning with the overall trend and further supporting our hypothesis that text-only training with X-REASONER improves multimodal performance.

Table 22: Performance on additional multimodal reasoning benchmarks.

| Model | V* (Wu & Xie, 2024) | EMMA (testmini) (Hao et al.) |
|---|---|---|
| Qwen-2.5-VL-7B | 45.5 | 25.5 |
| X-Reasoner | **49.8** | **27.2** |

## Q   ANALYSIS ON TEXT-ONLY TRAINING VS MULTI-MODAL TRAINING

We provide below a detailed analysis comparing multimodal and text-only training, which highlights the critical role of text-only training in eliciting multimodal reasoning. We further demonstrate the complementary benefits of combining both training paradigms.

### Q.1   TEXT-ONLY TRAINING IS MORE EFFECTIVE THAN MULTIMODAL TRAINING FOR ELICITING MULTIMODAL REASONING

In Table 23, we compare X-REASONER with multimodal finetuning strategy where we curate distilled CoT from GPT-4o on 100k medical PMC-VQA. We use the same prompt and the same filtering including rejection sampling for both text-based and multimodal data. We observe that X-REASONER with **text-only training** consistently improves over the baseline on multimodal reasoning tasks, whereas the gains from multimodal training alone are mixed. We find that while text-only X-REASONER improves performance on both reasoning and understanding subsets, multimodal training alone only improves the understanding portion and slightly degrades performance on the reasoning subset. Our hypothesis is that **existing multimodal datasets predominantly emphasize visual perception and short-form recognition, rather than multi-hop or long-chain reasoning**. As a result, multimodal data is less effective at inducing long-form chain-of-thought reasoning compared to carefully curated text-only reasoning data.

Table 23: Performance comparison on medical multimodal tasks across three training strategies

| Model | MedXpertQA-Reasoning | MedXpertQA-Understanding | NEJM |
|---|---|---|---|
| Qwen-2.5-VL-7B | 20.8 | 22.2 | 41.8 |
| X-Reasoner (text-only) | 22.3 | 27.4 | 45.7 |
| Multimodal | 20.5 | 23.1 | 37.2 |
| X-Reasoner + Multimodal | **29.3** | **31.0** | **50.8** |

To further validate this hypothesis, we analyze the output token length after RL training with text-only versus multimodal data (from Table 15 in the paper). On average, text-only RL produces substantially longer outputs (613 words) than multimodal RL (154 words). This large gap indicates that text-only RL is significantly more effective at inducing long and complex reasoning traces.

### Q.2   COMPLEMENTARY BENEFITS OF TEXT-ONLY AND MULTIMODAL TRAINING

Despite the above differences, we show that **text-only and multimodal training provide complementary strengths**, and their combination yields the best overall performance. Multimodal data contributes improved cross-modal alignment and visual grounding, while text-only data induces strong long-horizon reasoning. Their combination enables the model to leverage both capabilities.

To support this claim, we provide a detailed NEJM task error breakdown across three training strategies in Table 24. Out of 947 total examples, the cases where only X-Reasoner is correct (129) and where only multimodal training is correct (63) far exceed the cases where both methods are correct (51). This demonstrates that the two approaches exhibit minimal overlap and capture largely distinct subsets of solvable examples. Moreover, we observe a substantial number of examples (98) that are correctly solved only when both approaches are combined, i.e., using X-Reasoner + multimodal training. This further confirms the synergistic and complementary nature of text-only reasoning training and multimodal alignment training.

Table 24: NEJM error breakdown across training strategies.

| X-Reasoner | Multimodal Training | X-Reasoner + Multimodal Training | # Examples |
|---|---|---|---|
| correct | correct | correct | 203 |
| correct | correct | incorrect | 35 |
| correct | incorrect | correct | 129 |
| incorrect | correct | correct | 51 |
| correct | incorrect | incorrect | 71 |
| incorrect | correct | incorrect | 63 |
| incorrect | incorrect | correct | 98 |
| incorrect | incorrect | incorrect | 297 |

Below, we also present representative examples to illustrate the qualitative differences in reasoning traces produced by X-REASONER training, multimodal training, and X-REASONER + multimodal training. Overall, X-REASONER induces structured, step-by-step clinical reasoning and effectively integrates multimodal evidence. In contrast, multimodal training alone emphasizes detailed visual description but often exhibits simpler, less structured reasoning that can lead to incorrect conclusions. Notably, X-REASONER + multimodal training combines the strengths of both, preserving long, structured chains of thought while enriching them with fine-grained image analysis.

**X-REASONER (Correct).**

"Okay, let's try to figure out this question. The image shows a 12-year-old boy with recurrent infections. *(image description)*. Let's go through each option... Let me double-check... Given the options, CGD (C) is the most likely because recurrent infections are a hallmark of this condition. *(correct reasoning)*. The answer is CGD (C)."

**Multimodal Training (Incorrect).**

"The image shows a prominent swelling in the neck region, which suggests lymphadenopathy. *(detailed image description)* ... The prominent cervical lymphadenopathy seen in the image could be consistent with tuberculosis. *(incorrect reasoning)*. The answer is: $E$ : Tuberculous lymphadenitis."

**X-REASONER + Multimodal Training (Correct).**

"Let me review each option... The image provided shows a swollen lymph node, which could be consistent with CGD, where granulomas form. *(detailed image analysis with correct reasoning)* ... Checking the other options... Therefore, C: Chronic granulomatous disease is the most likely diagnosis."

## R    NOTES ON REPLICATING BASELINE RESULTS

The reported results for Qwen-2.5-VL-7B in our paper are based on our own replicated evaluations. We also observed a discrepancy between our replicated numbers and the originally reported

results, a phenomenon that has been independently noted by others in the open-source community. For example, the recent NeurIPS 2025 paper(Wang et al., 2025a) reports similar MMMU-val results and explicitly states that they cannot reproduce the official Qwen-2.5-VL-7B numbers using `lmms_eval`, `vlmevalkit`, or their native evaluation. Several hypotheses have been raised within the community to explain this discrepancy. These include potential performance regressions associated with recent versions of the Transformers library (Hugging Face Transformers Issue #40136) as well as differences in evaluation methodology—specifically, Qwen-2.5-VL-7B's original use of LLM-judge–based evaluation prompts that may be more tolerant of semantically similar answers rather than requiring exact matches (QwenLM/Qwen3-VL Issue #1149).

