# OpenReview forum: "X-REASONER: Towards Generalizable Reasoning Across Modalities and Domains"
_ICLR.cc/2026/Conference — Submitted to ICLR 2026_

### Official Review · Reviewer_gA6e · 2025-10-27

**Soundness:** 4
**Presentation:** 4
**Contribution:** 3
**Rating:** 6
**Confidence:** 4

**Summary:**

The authors build a two-stage text-only post-training pipeline on Qwen-2.5-VL-7B: first SFT using long CoT distillation, followed by reinforcement learning on math questions with verifiable rewards, yielding X-REASONER. Experiments show that this approach improves text tasks and, notably, transfers to multimodal benchmarks, surpassing several similarly sized models trained with multimodal supervision. Continuing text-only domain post-training in the medical vertical produces X-REASONER-MED, which achieves state-of-the-art results on multiple medical text and medical VLM benchmarks.

**Strengths:**

1. The paper proposes and validates a challenging hypothesis: text-only reasoning post-training in a general domain can induce cross-modality and cross-domain reasoning generalization, and in several cases outperforms direct multimodal/domain-specific supervision.
2. The paper is well structured; figures, tables, and writing are clear and easy to follow.
3. Without using multimodal supervision, the method attains strong performance on multimodal reasoning benchmarks such as MMMU, MMMU-Pro, and MathVista.

**Weaknesses:**

1. A key issue is the mismatch in data quality between the two compared training modes. For example, the OpenThoughts dataset is distilled from DeepSeek-R1, a very strong reasoning model, whereas some compared multimodal methods (e.g., MAmmoTH-VL2-7B) use training data that differ in text quality, quantity, and distribution. Such differences are likely decisive for performance and make the current comparisons unfair, casting doubt on the validity of the conclusions.
2. The reported results for Qwen-2.5-VL-7B are inconsistent with official numbers (e.g., official MMMU is 58.6, while this paper reports 53). Similar discrepancies appear for other methods and benchmarks. This inconsistency requires explanation, as it critically affects the strength of the experimental conclusions.
3. All results are based on Qwen-2.5-VL-7B. To strengthen the claim that “text-only post-training → cross-modality/cross-domain generalization,” key findings should be replicated on another 7B-class VL base model.
4. The conclusions would be more convincing with additional ablations and head-to-head controls, explicitly isolating the effects of data quality, data source.

**Questions:**

Same as the Weakness.

---

> ### Author Response · Authors · 2025-11-24
>
> Thank you for your review. Regarding weaknesses and questions.
>
> ## Weakness 1: Controlled study with multimodal and text-based training.
> To address this concern, we provide the following ablation studies on both the SFT and RL stages to fairly compare multimodal training and text-only training.
>
> For the SFT stage, we control the domain, quality and quantity of multimodal and text-only training settings to be as comparable as possible. For text-only data, we curate distilled CoT on 100k text-based medical data from DeepSeek-R1 (from MedQA, HeadQA and Med_MCQA). For multimodal data, since Deepseek-R1 is text-only, we choose GPT-4o to distill multimodal CoT traces on 100k medical PMC-VQA. We use the same prompt and the same filtering including rejection sampling for both text-based and multimodal data. Below, we show that text-only training significantly outperforms multimodal training on multimodal tasks.
> | Model                          | NEJM | MedXpertQA |
> |--------------------------------|----------------|------------|
> | Baseline                       | 41.8           | 21.4       |
> | text-only                      | 48.8           | 28.6       |
> | multimodal                     | 37.2           | 21.3       |
>
> For the RL stage, we also performed a controlled study as shown in Table 2 from the paper and also in below. The multimodal RL data comes from ThinkLite-VL-70k dataset which has slightly more data points than our text-only RL pipeline. Both datasets focus primarily on math-oriented questions, and neither includes distilled chain-of-thought supervision as the RL stage optimizes reasoning traces directly from the ground-truth labels, eliminating potential discrepancies that could arise from different teacher models. In this setting, we observe that our text-only recipe outperforms multimodal RL.
>
> | Method      | Data Quantity  | MathVision | MMMU-Pro |
> |-------------|------------------|-------|-------------|
> | Baseline    | -                | 24.7  | 38.3        |
> | text-only RL| 57k              | 28.1  | 41.2        |
> | Multimodal RL| 70k             | 25.9  | 40.1        |
>
> All of these controlled ablation studies show that our proposed text-only training can be even more effective than multimodal training under fair comparisons.
>
> ## Weakness 2: The reported results from Qwen-2.5-VL-7B are not consistent with what were reported in the original paper.
> The reported results for Qwen-2.5-VL-7B in our paper are based on our own replicated evaluations. We also observed a discrepancy between our replicated numbers and the originally reported results, which is a phenomenon that has been independently noted by others in the open-source community. For example, the recent NeurIPS 2025 paper **VL-Rethinker** ([TIGER-AI-Lab](https://github.com/TIGER-AI-Lab/VL-Rethinker)) reports similar MMMU-val results and explicitly states that they cannot reproduce the official Qwen-2.5-VL-7B numbers using lmms_eval, vlmevalkit, or their native evaluation.
>
> Several hypotheses have been raised within the community to explain this discrepancy. These include potential performance regressions associated with recent versions of the Transformers library ([Hugging Face Transformers Issue #40136](https://github.com/huggingface/transformers/issues/40136)) as well as differences in evaluation methodology (specifically, Qwen-2.5-VL-7B’s original use of LLM-judge–based evaluation prompts that may be more tolerant of semantically similar answers rather than requiring exact matches) ([QwenLM/Qwen3-VL Issue #1149](https://github.com/QwenLM/Qwen3-VL/issues/1149)).
>
> We will incorporate this explanation into the next revision of the paper.
>
> ## Weakness 3: Extending the findings to another model family/size
> Thanks for the suggestion. We provide below the generalization evaluation of the X-Reasoner recipe on a different model size (Qwen-2.5-VL-3B) and also on a different model family (GLM-4.1V-9B). We observe that the trend remains consistent across both model sizes and model families where the text-only X-Reasoner recipe consistently improves performance of the base model on all multimodal and domain-specific tasks.
>
> | Model                   | MMMU     | MMMU-Pro | MathVision (testmini) | MMMU-Pro-Health | MMMU-Health |
> |-------------------------|----------|----------|-----------------------|------------------|-------------|
> | Qwen-2.5-3B-VL          | 44.8     | 31.5     |  18.1                 |28.6             | 50.0        |
> | + X-Reasoner            | **51.0** | **35.8** | **30.3**              | **30.1**         | **52.6**    |
> | |
> | GLM-4.1V-9B            | 53.0      | 43.7     |    32.6                |    41.9       | 61.3           |
> | + X-Reasoner           | **60.2** | **48.2**   |  **35.5**             |  **48.9**        | **62.0** |
>
>
> ## Weakness 4: explicitly control for data source and data quality
> See rebuttal to weakness 1

---

### Official Review · Reviewer_BX1p · 2025-11-01

**Soundness:** 3
**Presentation:** 2
**Contribution:** 2
**Rating:** 4
**Confidence:** 4

**Summary:**

The paper proposes X-Reasoner, a vision-language model trained solely on general-domain, text-only data using a two-stage SFT–RL (GRPO) pipeline. Despite this training setup, the model exhibits strong cross-domain and cross-modal generalization (particularly to the vision modality). Building on this, the authors further train a medical-specific variant, X-REASONER-MED, which achieves state-of-the-art performance on medical benchmarks. The paper conducts extensive experiments to substantiate these claims.

**Strengths:**

1. The writing is clear and accessible, with a well-structured and logically organized presentation.
2. The experiments are reasonably solid and supported by rich implementation details.

**Weaknesses:**

1. The paper primarily explores whether a model trained solely on general text can generalize across domains and modalities. While this is an interesting direction, I have some reservations about relying exclusively on text-only training for a VLM. From an architectural perspective, this setup may under-emphasize vision–language alignment and mainly benefit the LLM backbone. In parallel, recent work suggests that directly training on multimodal data is also an effective and perhaps more targeted path—for example, VL-Rethinker [1] and MiMo-VL [2] report strong gains.
2. Regarding scope, the cross-domain generalization claim is mainly evaluated in the medical domain. Expanding to additional domains would help solidify the conclusion.
3. On evaluation coverage, several recent baselines and benchmarks appear to be missing. It would strengthen the paper to include baselines such as VL-Rethinker-7B[1], Mimo-VL[2], ThinkLite-VL-7B[3], LLaVA-Critic-R1[4] and so on,  and to expand benchmarks (e.g., V*[5], “VLMs Are Blinds”[6], BLINK[7], EMMA[8], etc.).
4.The ablation in Takeaway 3.2.2, probing whether the model truly relies on visual information, is interesting but could be further strengthened. The filtering procedure and the filtering model might benefit from refinement [8]. In addition, evaluating on vision-centric reasoning suites (e.g., V* and “VLMs Are Blind”) could further corroborate the findings.

[1] https://arxiv.org/pdf/2504.08837
[2] https://arxiv.org/pdf/2506.03569
[3] https://arxiv.org/abs/2504.07934
[4] https://arxiv.org/pdf/2509.00676
[5] https://arxiv.org/abs/2312.14135
[6] https://arxiv.org/abs/2407.06581
[7] https://arxiv.org/abs/2404.12390
[8] https://arxiv.org/abs/2501.05444

**Questions:**

Please see weaknesses

---

> ### Author Response · Authors · 2025-11-24
>
> Thank you for your review. Regarding weaknesses and questions.
>
> ## Weakness 1: Emphasize on LLM backbone rather than multimodal alignment
> Our central motivation is to understand how much of the multimodal reasoning improvement can be attributed purely to text-only training i.e. from the LLM backbone. Additionally, since we are performing reasoning training from a VLM model that is already aligned in the multimodal space, we can investigate whether the additional language model reasoning post-training can be combined to the existing multimodal alignment from the base model. Our paper confirms this hypothesis.
>
> In addition, we provide another experiment below to show that x-reasoner can be extended to multimodal setting and can serve as a good foundation. We provide below an ablation study to show that adding multimodal data (we curated 100k multimodal PMC-VQA data and distilled reasoning traces from GPT-4o) can further improve x-reasoner's performance on multimodal tasks.
>
> | Model                            | NEJM | MedXpertQA |
> |----------------------------------|------|------------|
> | Qwen-2.5-VL-7B                   | 41.8 | 21.4       |
> | X-Reasoner                       | 45.7 | 23.8       |
> | X-Reasoner + Multimodal Training | 50.8 | 29.8       |
>
> ## Weakness 2: The scope of cross-domain generalization beyond medical domain
> Thanks for the feedback. We further evaluate X-Reasoner on two additional domains: history domain from MMLU-History and finance domain from MMMU-Pro-Finance and we have included results below. We show that X-Reasoner's cross-domain generalization extends beyond medical domain to other specialized domains as demonstrated by the substantial gain from X-Reasoner on History and Finance domains.
> | Model                | History domain         | Finance domain                  |
> |----------------------|------------------------|---------------------------------|
> | Qwen-2.5-VL-7B       | 42.7                  | 45.3                     |
> | X-Reasoner           | **44.6**                  | **50.0**                     |
>
> ## Weakness 3: Other more recent baselines.
> Below is a comparison between X-Reasoner and the requested baseline models that are also built on top of Qwen-2.5-VL-7B. We observe that X-Reasoner performs competitively with these more recent and stronger multimodal reasoning, and in fact still achieves state-of-the-art performance across both general-domain and medical domain multimodal datasets.
>
> | Model              | MMMU | MMMU-Pro | MMMU-H | NEJM |
> |--------------------|------|----------|--------|------|
> | Qwen baseline      | 53.0 | 38.7     | 55.5   | 41.8 |
> | X-Reasoner         | **56.4** | **43.0** | **62.9** | **45.7** |
> | VL-Rethinker-7B    | 56.3 | 41.7     | 60.7   | 43.5 |
> | ThinkLite-VL-7B    | 55.5 | 39.0     | 62.0   | 45.0   |
>
>
> ## Weakness 4: Expanding evaluation to include vision-centric tasks
> Below we report X-Reasoner’s performance on additional requested benchmarks. X-Reasoner consistently surpasses the baseline model, aligning with the overall trend and supporting our hypothesis from the study that text-only training from X-Reasoner improves multimodal performance.
>
> | Model             | V\_star | EMMA (testmini) |
> |-------------------|---------|-----------|
> | Qwen-2.5-VL-7B    | 45.5    | 25.5      |
> | X-Reasoner        | **49.8** | **27.2** |

---

> > ### Comment · Reviewer_BX1p · 2025-11-27
> >
> > Thank you very much for your detailed rebuttal and for running the additional experiments. I appreciate the effort in extending the evaluations to more domains and benchmarks, as well as the new multimodal training ablations.
> >
> > I believe the paper would be significantly strengthened by a deeper comparative analysis regarding the specific gains offered by pure text training versus multimodal training. For instance, it would be valuable to investigate why fine-tuning the base model directly on 100k multimodal PMC-VQA data leads to a performance drop on NEJM, whereas training on the x-reasoner leads to performance improvements and training on text-only data leads to improvements as well.
> >
> > I would be very interested to see more such analysis in a future revision. For the current version, I will keep my original score.

---

> > > ### Author Response · Authors · 2025-12-01
> > >
> > > We thank the reviewer for the feedback. In response, we provide below a detailed analysis comparing multimodal and text-only training, which highlights the critical role of text-only training in eliciting multimodal reasoning. We further demonstrate the complementary benefits of combining both training paradigms.
> > >
> > > ---
> > >
> > > ## 1. Text-only training is more effective than multimodal training for eliciting multimodal reasoning
> > >
> > > We observe that X-Reasoner with **text-only training** consistently improves over the baseline on multimodal reasoning tasks, whereas the gains from multimodal training alone are mixed. To better understand this behavior, we conduct a finer-grained analysis on the MedXpertQA benchmark, which can be divided into **understanding** and **reasoning** subsets.
> > >
> > > We re-ran our experiments comparing X-Reasoner trained with text-only data versus multimodal fine-tuning using 100k PMC-VQA samples. The results are summarized below:
> > >
> > > | Model                   | MedXpertQA-Reasoning | MedXpertQA-Understanding |
> > > |-------------------------|----------------------|--------------------------|
> > > | Baseline                | 20.8                 | 22.2                     |
> > > | X-Reasoner (text-only)  | 22.3                 | 27.4                     |
> > > | Multimodal              | 20.5                 | 23.1                     |
> > > | X-Reasoner + Multimodal | **29.3**             | **31.0**                 |
> > >
> > > We find that while text-only X-Reasoner improves performance on both reasoning and understanding subsets, multimodal training alone only improves the understanding portion and slightly degrades performance on the reasoning subset.
> > >
> > > Our hypothesis is that **existing multimodal datasets predominantly emphasize visual perception and short-form recognition, rather than multi-hop or long-chain reasoning**. As a result, multimodal data is less effective at inducing long-form chain-of-thought reasoning compared to carefully curated text-only reasoning data.
> > >
> > > To further validate this hypothesis, we analyze the output token length after RL training with text-only versus multimodal data (from Table 2 in the paper). On average, text-only RL produces substantially longer outputs (613 words) than multimodal RL (154 words). This large gap indicates that text-only RL is significantly more effective at inducing long and complex reasoning traces.
> > >
> > > ---
> > >
> > > ## 2. Complementary benefits of text-only and multimodal training
> > >
> > > Despite the above differences, we emphasize that **text-only and multimodal training provide complementary strengths**, and their combination yields the best overall performance. Multimodal data contributes improved cross-modal alignment and visual grounding, while text-only data induces strong long-horizon reasoning. Their combination enables the model to leverage both capabilities.
> > >
> > > To support this claim, we provide a detailed NEJM error breakdown across three training strategies:
> > >
> > > | X-Reasoner | Multimodal Training | X-Reasoner + Multimodal Training | # Examples |
> > > |------------|---------------------|----------------------------------|------------|
> > > | correct         | correct                  | correct                              | 203        |
> > > | correct       | correct                | incorrect                               | 35         |
> > > | correct         | incorrect                 | correct                              | 129        |
> > > | incorrect         | correct                 | correct                             | 51         |
> > > | correct         | incorrect                 | incorrect                              | 71         |
> > > | incorrect        | correct             | incorrect                               | 63         |
> > > | incorrect         | incorrect                 | correct                              | 98         |
> > > | incorrect        | incorrect                 | incorrect                               | 297        |
> > >
> > > Out of 947 total examples, the cases where only X-Reasoner is correct (129) and where only multimodal training is correct (63) far exceed the cases where both methods are correct (51). This demonstrates that the two approaches exhibit minimal overlap and capture largely distinct subsets of solvable examples.
> > >
> > > Moreover, we observe a substantial number of examples (98) that are correctly solved only when both approaches are combined, i.e., using X-Reasoner + multimodal training. This further confirms the synergistic and complementary nature of text-only reasoning training and multimodal alignment training.

---

> > > > ### Author Response · Authors · 2025-12-03
> > > >
> > > > In continuation to the last comment, we also present representative examples to illustrate the qualitative differences in reasoning traces produced by X-Reasoner training, multimodal training, and X-Reasoner + multimodal training. Overall, X-Reasoner induces structured, step-by-step clinical reasoning and effectively integrates multimodal evidence. In contrast, multimodal training alone emphasizes detailed visual description but often exhibits simpler, less structured reasoning that can lead to incorrect conclusions. Notably, X-Reasoner + multimodal training combines the strengths of both, preserving long, structured chains of thought while enriching them with fine-grained image analysis.
> > > >
> > > > ### X-Reasoner (Correct)
> > > > > “Okay, let’s try to figure out this question. The image shows a 12-year-old boy with recurrent infections. *(image description)*. Let’s go through each option… Let me double-check… Given the options, CGD (C) is the most likely because recurrent infections are a hallmark of this condition. *(correct reasoning)*. The answer is CGD (C).”
> > > >
> > > > ### Multimodal Training (Incorrect)
> > > > > “The image shows a prominent swelling in the neck region, which suggests lymphadenopathy. *(detailed image description)* … The prominent cervical lymphadenopathy seen in the image could be consistent with tuberculosis. *(incorrect reasoning)*. The answer is: \(E: \text{Tuberculous lymphadenitis}\).”
> > > >
> > > > ### X-Reasoner + Multimodal Training (Correct)
> > > > > “Let me review each option… The image provided shows a swollen lymph node, which could be consistent with CGD, where granulomas form. *(detailed image analysis with correct reasoning)* … Checking the other options… Therefore, C: Chronic granulomatous disease is the most likely diagnosis.”

---

### Official Review · Reviewer_5fph · 2025-11-03

**Soundness:** 3
**Presentation:** 3
**Contribution:** 3
**Rating:** 6
**Confidence:** 2

**Summary:**

This paper introduces X-REASONER, a vision-language model designed to enhance generalizable reasoning abilities across both modalities and domains. It explores whether reasoning capabilities can be generalized across these dimensions and demonstrates that training based on general-domain text is effective for reasoning generalization. The model is trained with a novel two-stage strategy and performs well in various tasks on domain-specific multimodal data, including complex medical reasoning benchmarks.

**Strengths:**

1. The paper presents a novel approach to generalizable reasoning, showing that general-domain text-based training can effectively improve reasoning capabilities across domains and modalities.Being trained only on text-based data, the model surpass other MLLMs with similar model size.
2. The model is trained with a novel two-stage strategy, combined with SFT+CoT and R, and author give detailed experiments results to demonstrate how to come up with such a strategy.

**Weaknesses:**

1. The models are only trained on 7B. It may be helpful to show robustness of the training strategy with some smller/bigger model size.
2. Some metrics in figures (like pass@5 in Fig 3 and 4) is not well introduced, which makes it complicated to understand the performance of the models in the first glance.

**Questions:**

1. The X-REASONER is trained on text-only data and get good results. I'd like to know whether this training strategy can be extended to multimodal datasets, which may further improve the model's ability.
2. While larger model size cost too much computation resources, can experiments on small model size (like 3B) be carried out to show the robustness of the method?
3. In Fig.4 the origin X-REASONER outperforms X-REASONER-Med on pass@5 metrics. Can authors explain the reasons in detail?

---

> ### Author Response · Authors · 2025-11-24
>
> Thank you for your review. Regarding the weaknesses and questions.
> ## weakness 1: generalization to other model sizes
> Thanks for the suggestion. We provide below the generalization evaluation of the X-Reasoner recipe on a different model size (Qwen-2.5-VL-3B) and also on a different model family (GLM-4.1V-9B). We observe that the trend remains consistent across both model sizes and model families where the text-only X-Reasoner recipe consistently improves performance of the base model on all multimodal and domain-specific tasks.
>
> | Model                   | MMMU     | MMMU-Pro | MathVision (testmini) | MMMU-Pro-Health | MMMU-Health |
> |-------------------------|----------|----------|-----------|------------------|-------------|
> | Qwen-2.5-3B-VL          | 44.8     | 31.5     |  18.1  |28.6             | 50.0        |
> | + X-Reasoner         | **51.0** | **35.8** | **30.3**  | **30.1**         | **52.6**    |
> |                   |      |          |                        |                  |             |
> | GLM-4.1V-9B        | 53.0   | 43.7     |    32.6  |    41.9     | 61.3            |
> | + X-Reasoner           | **60.2** | **48.2**   |  **35.5**  |  **48.9**        | **62.0** |
>
> ## weakness 2: Metrics are not well introduced
> Thanks for raising the issue. We will further clarify the metric definition in the Evaluation Setup paragraph in section 3: "we report average accuracy, majority-vote accuracy, and pass@n (counting an example correct if any of n attempts is correct) accuracy over five runs at temperature 0.3 to ensure robustness and reproducibility."
>
> ## Question 1: Can x-reasoner training be extended to multimodal setting?
> Yes, x-reasoner can be extended to multimodal setting and can serve as a good foundation. We provide below an ablation study to show that adding multimodal data (we curated 100k multimodal data (from PMC-VQA) and distilled reasoning traces from GPT-4o) can further improve X-reasoner's performance on multimodal tasks.
>
> | Model                            | NEJM | MedXpertQA |
> |----------------------------------|------|------------|
> | Baseline (Qwen-2.5-VL-7B)        | 41.8 | 21.4       |
> | X-Reasoner                       | 45.7 | 23.8       |
> | X-Reasoner + Multimodal Training | 50.8 | 29.8       |
>
> ## Question 2: smaller model to test robustness
> We addressed this concern in weakness 1
>
> ## Question 3: X-Reasoner outperforms X-Reasoner-Med on pass@5 accuracy in Figure 4
> Yes, we also observe this behavior. We have the following analysis in section 3.2:
> "Notably, X-REASONER sometimes attains higher pass@n scores, suggesting it explores a broader search space. Conversely, X-REASONER-MED, benefiting from targeted medical-domain fine-tuning, already leverages this search space more effectively but potentially with reduced room for further gains. "
>
> To elaborate it a bit further, since pass@5 can be viewed as a recall-like metric that reflects how effectively a model explores its sample space, this behavior suggests a distributional shift after the additional medical RL stage from X-Reasoner-Med. Our interpretation is that X-Reasoner-Med has converted that exploratory potential into a sharper, more concentrated output distribution, yielding more precise and consistently correct primary answers—an attribute that is typically more valuable in real-world settings where a single definitive answer is required. In contrast, X-Reasoner remains more exploratory, producing a wider range of candidate answers that improves recall (pass@5) but reduces precision.

---

### Author Response · Authors · 2025-11-24
**General Rebuttal**

We thank all reviewers for carefully reading our paper and providing constructive feedback and suggestions. We greatly appreciate the recognition that the paper is well written and that it introduces a novel approach and training strategy supported by solid experiments validating an important and challenging hypothesis: that text-only reasoning training can yield strong cross-domain and cross-modality generalization.

We have addressed all reviewer concerns as thoroughly as possible in the rebuttal and have updated the paper accordingly (the revisions are highlighted in blue). Specifically:

### **Generalization of the X-Reasoner recipe**

We conducted additional experiments applying the X-Reasoner recipe to different model size (Qwen-2.5-VL-3B) and to different model family (GLM-4.1V-9B). We observed the same trend, i.e. text-only reasoning training consistently improves over the baseline on cross-domain and cross-modality benchmarks, and the X-Reasoner recipe proves to be generalizable.

### **Controlled study on text-only vs multimodal training**

We added detailed analysis comparing text-only and multimodal-training to further confirm our hypothesis that text-only reasoning training can be equally or even more effective than multimodal training. We further demonstrate that X-Reasoner can be extended to multimodal fine-tuning for further gains. We provide both quantitative error breakdowns and qualitative examples to facilitate our analysis.

### **Additional evaluation benchmarks, domains and baselines**

We extended our evaluation to additional domains (e.g., history, finance), newer benchmarks (V_star, EMMA), and stronger recent baselines (VL-Rethinker, ThinkLite-VL). Across all added settings, X-Reasoner maintains consistent improvements and remains competitive with the recent models.

### **Clarification and explanations**
We also provided clarifications and detailed explanations to address all remaining reviewer's questions and concerns.

---

### Meta-Review · Area_Chair_X81M · 2026-01-08

**Summary:**

This paper introduces the interesting finding that general domain test-based post-training can enable strong generalization in general-domain and modality reasoning. It introduces X-REASONER, a vision-language model with reasoning post-training solely from general-domain text for generalizable reasoning. It receives the initial scores of 2 borderline accept and 1 borderine reject. Reviewers raised concerns on the generalization of X-Reasoner to different model sizes, domains, and modalities, text-only v.s. multimodal training, and experimental validation.

**Reviewer Concerns:**

Some concerns are address during the rebuttal with additional experimental results, e.g., on difference size models, different domains, and different base models. Reviewer BX1p raised a concern on relying on test-only training for VLMs. Reviewer believes that more in-depth analysis and insights on multimodal training v.s. text-only training in a future version will strengthen the paper. While the authors provide some experimental analysis in the subsequent rebuttal, they only propose a hypothesis to the phenomenon. The area chair believes that more rigorous analysis and study is needed for this problem.

**Reviewer Scores:**

Two reviewers are likely to keep their borderline accept scores. While the other reviewer, BX1p, is not fully convinced by the rebuttal. He/She mentioned that he/she will keep the original score of borderline reject for the current version. The area chair shares the concern of reviewer BX1p on the insufficient study on multimodal v.s. test-only training. The current version with only a hypothesis is not rigorous and solid enough.

---

### Decision · Program_Chairs · 2026-01-26

Reject